# Towards Efficient and Expressive GNNs for Graph Classification via Subgraph-aware Weisfeiler-Lehman

**Zhaohui Wang**
Institute of Computing Technology, CAS
University of Chinese Academy of Sciences
wangzhaohui18b@ict.ac.cn

**Qi Cao**
Institute of Computing Technology, CAS
caoqi@ict.ac.cn

**Huawei Shen** *
Institute of Computing Technology, CAS
University of Chinese Academy of Sciences
shenhuawei@ict.ac.cn

**Bingbing Xu**
Institute of Computing Technology, CAS
xubingbing@ict.ac.cn

**Muhan Zhang** *
Institute for Artificial Intelligence, Peking University
muhan@pku.edu.cn

**Xueqi Cheng**
Institute of Computing Technology, CAS
University of Chinese Academy of Sciences
cxq@ict.ac.cn

## Abstract

The expressive power of GNNs is upper-bounded by the Weisfeiler-Lehman (WL) test. To achieve GNNs with high expressiveness, researchers resort to subgraph-based GNNs (WL/GNN on subgraphs), deploying GNNs on subgraphs centered around each node to encode subgraphs instead of rooted subtrees like WL. However, deploying multiple GNNs on subgraphs suffers from much higher computational cost than deploying a single GNN on the whole graph, limiting its application to large-size graphs. In this paper, we propose a novel paradigm, namely Subgraph-aware WL (SaWL), to obtain graph representation that reaches subgraph-level expressiveness with a single GNN. We prove that SaWL has beyond-WL capability for graph isomorphism testing, while sharing similar runtime to WL. To generalize SaWL to graphs with continuous node features, we propose a neural version named Subgraph-aware GNN (SaGNN) to learn graph representation. Both SaWL and SaGNN are more expressive than 1-WL while having similar computational cost to 1-WL/GNN, without causing much higher complexity like other more expressive GNNs. Experimental results on several benchmark datasets demonstrate that fast SaWL and SaGNN significantly outperform competitive baseline methods on the task of graph classification, while achieving high efficiency.

## 1 Introduction

Graph-structured data widely exist in the real world, and modeling graphs has become an important topic in the field of machine learning. Graph learning has widespread applications [1–3], and many valuable applications can be formulated as graph classification, e.g., molecular property prediction [4], drug toxicity prediction [5]. Graph classification aims to predict the label of the given graph by exploiting graph structure and feature information. Learning expressive representations of graphs is crucial for classifying graphs of different structural characteristics.

---

*Corresponding authors

Z. Wang et al., Towards Efficient and Expressive GNNs for Graph Classification via Subgraph-aware Weisfeiler-Lehman. *Proceedings of the First Learning on Graphs Conference (LoG 2022)*, PMLR 198, Virtual Event, December 9–12, 2022.

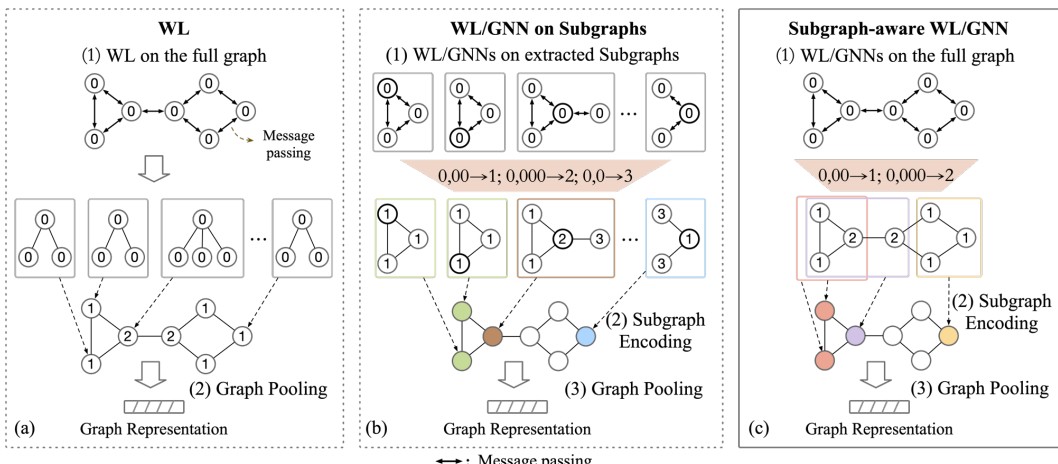

**Figure 1:** (a) WL encodes nodes by rooted subtrees, which has limited expressiveness. (b) WL/GNN on Subgraphs paradigm extracts rooted subgraphs and applies GNNs on each rooted subgraph, which is computationally expensive. (c) Our Subgraph-aware WL/GNN applies WL/GNN on the full graph and then encodes rooted subgraphs by aggregating nodes within the subgraph. The proposed paradigm possesses higher expressive power than 1-WL while keeping the computational cost low.

Recently, Graph Neural Networks (GNNs) have achieved great success in graph classification tasks [6–8]. GNNs that follow a message passing scheme first iteratively aggregate neighbor information to update node representations, then pool node representations into graph-level representations [9]. Essentially, GNNs are parameterized generalizations of the 1-dimensional Weisfeiler-Lehman algorithm (1-WL) [10], which encodes each node by its rooted subtree pattern [11], as shown in Figure 1 (a). Despite the success of traditional message passing GNNs, the expressive power of GNNs is theoretically upper-bounded by 1-WL, which is known to have limited power in distinguishing many non-isomorphic graphs [12–14].

To uplift the expressive power of GNNs, researchers adopt a paradigm of *WL/GNN on subgraphs* (Figure 1 (b)), which encodes rooted subgraphs instead of rooted subtrees as node representations [15–17]. Methods under the paradigm first extract rooted subgraphs (i.e., subgraph induced by the neighbor nodes within $h$ hops of a center node), and then apply GNNs on each extracted subgraph respectively. However, as GNNs are applied to subgraphs extracted from each node of the graph, the computational cost of these methods is much higher than that of traditional message passing GNNs, especially when the subgraphs have similar sizes to the whole graph.

In this paper, we propose a novel paradigm of *Subgraph-aware WL/GNN* (SaWL), which reaches higher expressiveness than 1-WL with a single GNN (Figure 1 (c)). It first deploys WL/GNN on the full graph to obtain node representations, and then aggregates the nodes within each subgraph to achieve subgraph awareness. The proposed paradigm greatly reduces the computational cost of existing WL-on-subgraph methods, while achieving higher expressive power than 1-WL. Under the paradigm, we propose an algorithm as fast implementation of SaWL, which consists of a WL encoder and a subgraph operator ($S$ operator). We first apply a standard WL on the full graph to iteratively update each node label based on its current label and the labels of its neighbors [18]. After $h$-th iteration of WL, we use the $S$ operator to encode the $h$-hop rooted subgraph of each node by aggregating the current labels of nodes within the subgraph. The whole graph feature mapping at this iteration is obtained further by pooling the subgraph feature mapping. Finally, we concatenate graph feature mappings at different iterations into a final graph feature mapping for graph classification. We then generalize SaWL to a neural version, Subgraph-aware GNN (SaGNN).

Compared to the paradigm of *WL/GNN-on-subgraphs*, the proposed *Subgraph-aware WL/GNN* does not need to copy a full $n$-node graph into $n$ subgraphs (each rooted at a node) and run WL/GNN on each subgraph separately (thus the same node can have multiple representations when appearing in different subgraphs). Instead, Subgraph-aware WL/GNN only runs WL/GNN on the full graph and encodes subgraph information based on the "global" WL/GNN node representations. It encodes

the subgraph information while avoiding the need to apply WL/GNN on each extracted subgraph respectively, which improves the expressiveness and keeps low computational cost at the same time.

We evaluate the effectiveness of the proposed fast SaWL and SaGNN on graph classification tasks via several benchmark datasets. We then conduct the expressive power evaluation and running time comparison to verify the high effectiveness and high efficiency of our methods.

## 2 Preliminary

### 2.1 Weisfeiler-Lehman and Feature Mapping

***Weisfeiler-Lehman*** (1-WL) [10] is one of the most widely used algorithms which can tackle graph isomorphism testing for a broad class of graphs [19, 20]. Specifically, 1-WL proceeds in iterations denoted by $h$, and each iteration includes multisets determination, injective mapping and relabeling [18]. Given two graphs $G$ and $H$, firstly, WL aggregates the labels of neighbor nodes as a multiset $M_v^h$. For $h = 0$, $M_v^0 = l_v^0$, and for $h > 0$, $M_v^h = \{\!\{l_u^{h-1} | u \in \mathcal{N}(v)\}\!\}$, where $l_v^h$ is the label of node $v$ in the $h$-th iteration, $\mathcal{N}(v)$ denotes the neighbor nodes of $v$ and $\{\!\{\}\!\}$ denotes a multiset. Note that multiset is a generalized set that allows repeated elements [13]. Then, an injective function is required to update the label of node, $l_v^h := \text{HASH}\left(\left(l_v^{h-1}, M_v^h\right)\right)$. The procedures repeat until the multisets of node labels of two graphs differ, the number of iterations reaches a predetermined value, or the node labels do not change in one iteration. ***The feature mapping*** of the whole graph can be obtained by mapping multisets after each iteration. Although 1-WL works well in testing isomorphism on many graphs, the distinguishing power of the 1-WL is limited [12, 21].

### 2.2 Graph Neural Networks

Traditional message passing Graph Neural Networks (GNNs) follow an aggregation and update scheme, which can be viewed as the neural implementation of the 1-WL [13, 22]. Nodes aggregate features of neighbor nodes, combine them with its features and update to new representations:

$$\boldsymbol{h}_v^k = \text{UPDATE}\left(\boldsymbol{h}_v^{k-1}, \text{AGGREGATE}\left(\boldsymbol{h}_u^{k-1} | u \in \mathcal{N}(v)\right)\right), \tag{1}$$

where the UPDATE and AGGREGATE functions are implemented with neural networks. Then, the whole graph representation can be computed by a pooling/readout operation like sum [23–25]:

$$\boldsymbol{h}^k(G) = \text{READOUT}\left(\boldsymbol{h}_v^k | v \in \mathcal{V}(G)\right). \tag{2}$$

GNNs have been popular architectures for representation learning on graphs. However, it has been proved that the expressive power of message passing GNNs is upper bounded by the 1-WL algorithm [13, 14], which limits the performance on graph classification tasks.

## 3 Subgraph-aware Weisfeiler-Lehman

We propose a new paradigm of Subgraph-aware Weisfeiler-Lehman (SaWL), which exceeds the expressive power of 1-WL while keeping low computational complexity. The paradigm first iteratively applies WL/GNN to the original input graph. With the obtained node representations at each iteration, the paradigm encodes each rooted subgraph by hashing the node representations within its range. Then, the subgraph representations are pooled to obtain the whole graph representation.

### 3.1 SaWL for Graph Classification

SaWL consists of a WL encoder, a subgraph encoding operator (the $S$ operator) and a graph feature mapping module. For graph $G$, the **WL encoder** executes normal WL steps described in section 2.1, which outputs the updated node labels $\{l_v^h | v \in \mathcal{V}(G)\}$, where $l_v^h$ is the label of node $v$ in the $h$-th iteration. The core of the proposed SaWL lies in the additional $S$ operator, which encodes subgraph information with the results of each WL iteration. We describe the $S$ operator in the following.

$S$ **operator.** We employ an injective hash function that acts on labels of nodes within the subgraph to encode the subgraph information into a subgraph feature mapping:

$$\phi^{(h)}\left(G_v^h\right) = \text{HASH}\left(\{\!\{l_v^h | v \in \mathcal{V}(G_v^h)\}\!\}\right), \tag{3}$$

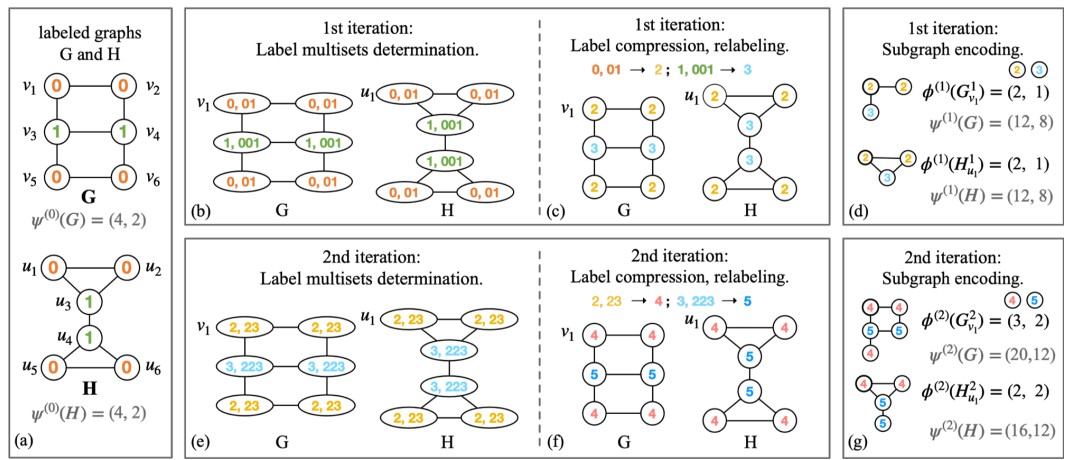

**Figure 2:** Illustration of the fast SaWL. Colored numbers denote node labels. In (b), (c), (e) and (f), neighbor nodes are aggregated as multiset and compressed to updated labels (the same as 1-WL). In (d) and (g), the $S$ operator encodes each rooted subgraph into a feature mapping. After the 2nd iteration, the feature mapping of $G_{v_1}^2$ is no longer equal to that of $H_{u_1}^2$, so that graph $G$ and $H$ can be discriminated by SaWL (but not by 1-WL).

where $G_v^h$ is the $h$-hop rooted subgraph around node $v$. The hash function can be designed freely. Essentially, the $S$ operator encodes the multiset of node labels within $G_v^h$ (obtained by running $h$ iterations of WL on the full graph) into a subgraph representation.

**Graph Feature Mapping Module.** With the subgraph feature mapping, an injective readout function is adopted to obtain the whole graph feature mapping in the $h$-th iteration, i.e.,

$$\boldsymbol{\psi}^{(h)}(G) = \text{READOUT}\left(\boldsymbol{\phi}^{(h)}(G_v^h)|v \in \mathcal{V}(G)\right). \tag{4}$$

The readout function can be chosen freely. To retain the structural information at all iterations, the final graph feature mapping is obtained by concatenation, i.e., $\boldsymbol{\psi}(G) = \text{CONCAT}\left(\boldsymbol{\psi}^{(0)}(G), \boldsymbol{\psi}^{(1)}(G), ..., \boldsymbol{\psi}^{(H)}(G)\right)$, where $H$ is the maximum iteration number.

**Discussion.** Compared to plain WL, which directly uses node labels at $h$-th iteration to obtain the graph representation, SaWL additionally uses the multiset of labels of node $v$'s neighbors within $h$-hop to enhance WL with subgraph information. To understand SaWL's benefits over plain WL, from one point of view, SaWL encodes the node-subgraph-graph hierarchy instead of the node-graph hierarchy of WL, which better captures the hierarchical structural characteristics of the graph. From another point of view, plain WL encodes a node by its rooted subtree pattern, which can have repeated nodes. The repetitions of the same node are regarded as distinct nodes, and the actual number of nodes in the subtree pattern might be corrupted. The hash function in the $S$ operator further characterizes the information of the actual number of nodes in the subgraph (which also equals the actual number of nodes in the subtree pattern, because the subgraph $G_v^h$ does not have repeated nodes).

### 3.2 A Fast Implementation of SaWL

To illustrate the idea of SaWL, we provide a particular implementation here named **fast SaWL**. For the $S$ operator, we design HASH function as a counting mapping that counts the occurrence of different node labels in the subgraph. Then, we adopt sum pooling as the READOUT function to obtain the whole graph feature mapping.

**Definition 1 (Counting mapping).** *Let $\mathcal{L}_h \subseteq \mathcal{L}$ denote the set of node labels that occur at least once in the $h$-th iteration. $\mathcal{L}_h = \{\ell_1^h, \ell_2^h, ..., \ell_{|\mathcal{L}_h|}^h\}$ and we assume that $\mathcal{L}_h$ is ordered. Assume $G_v^h \in \mathcal{G}$, where $\mathcal{G}$ is the complete graph space. For each iteration $h$, we define a counting mapping*

$c_h : \mathcal{G} \times \mathcal{L}_h \to \mathbb{N}$, where $c_h(G_v^h, \ell_i^h)$ is the number of the occurrences of the $i$-th node label $\ell_i^h$ in subgraph $G_v^h$ at the $h$-th iteration.

With counting mapping, the feature mapping of the subgraph $G_v^h$ can be obtained by $\phi^{(h)}(G_v^h) = \left( c_h(G_v^h, \ell_1^h), ..., c_h(G_v^h, \ell_{|\mathcal{L}_h|}^h) \right)$, where the value of the $i$-th position of the vector represents the occurrence number of label $\ell_i^h$ in the $h$-th iteration. Essentially, the $S$ operator encodes subgraph by mapping the multiset of node labels within the subgraph to a vector, recording the occurrence number of each label. Then, the whole graph feature mapping is obtained by applying sum pooling to the subgraph feature mappings. Although the sum pooling is not an injective readout function, as we will show, it allows fast computation (acceleration) via an implementation trick.

**Illustration.**   We illustrate the fast SaWL in Figure 2. Given two graphs $G$ and $H$ where colored numbers indicate node labels. The WL encoder of fast SaWL updates node labels in (b), (c), (e) and (f). $S$ operator encodes rooted subgraphs, and we take two rooted subgraphs as examples in Figure 2(g). The feature mapping of the subgraph $G_{v_1}^2$ in the 2nd iteration is $\phi^{(2)}(G_{v_1}^2) = (3, 2)$, which means the label 4 occurs three times and label 5 occurs twice in the subgraph. Then the subgraphs are pooled to obtain the graph feature mapping in the 2nd iteration, e.g., for graph $G$, $\psi^{(2)}(G) = \phi^{(2)}\left(G_{v_1}^2\right) + \phi^{(2)}\left(G_{v_2}^2\right) + ... + \phi^{(2)}\left(G_{v_6}^2\right) = (20, 12)$. And for graph $H$, $\psi^{(2)}(H) = (16, 12)$. Finally, the whole graph feature mappings are $\psi(G) = (4, 2, 12, 8, 20, 12)$, and $\psi(H) = (4, 2, 12, 8, 16, 12)$. The graph $G$ and $H$ cannot be discriminated by 1-WL, but they can be discriminated by our fast SaWL.

**Acceleration.**   In fast SaWL, the calculation of the $S$ operator can be executed simultaneously with the WL encoder, which reduces the computational time. Since the subgraph feature mappings are summed as the whole graph feature mapping, the frequency of one node contributing to the whole graph feature mapping is equal to the number of occurrences of this node in all $h$-hop rooted subgraphs. We use graph $H$ (adapted from Figure 2(f)) as an example. In Figure 3(a), each tuple $(a, b)$ represents the feature mapping of the node's rooted subgraph. The whole graph feature mapping can be computed by summing all subgraphs' feature mappings: $\psi^{(2)}(H) = (2, 2) + ... + (4, 2) + ... + (2, 2) = (16, 12)$. However, we can actually compute the whole graph feature mapping from a global perspective. E.g., node $u_1$ contributes to the 2-hop rooted subgraphs of nodes $u_1, u_2, u_3, u_4$. And the

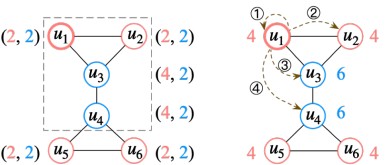

(a) Subgraph-perspective    (b) Global graph-perspective

**Figure 3:** $u_1$ contributes to the feature mappings of rooted subgraphs of $u_1, u_2, u_3, u_4$. The contribution number equals the size of rooted subgraph $H_{u_1}^{(2)}$.

number of $u_1$'s contributions to the whole graph feature mapping is exactly the size of node $u_1$'s 2-hop rooted subgraph, i.e., $|\mathcal{V}(H_{u_1}^{(2)})| = 4$. Similarly, we mark each node's contribution number beside it in Figure 3(b). The whole graph feature mapping can be alternatively computed by summing the contribution numbers for each label dimension, i.e., $\psi^{(2)}(H) = (4 + 4 + 4 + 4, 6 + 6) = (16, 12)$. The sizes of rooted subgraphs can be computed together in the multiset determination of WL running on the original graph by propagating node label and ID simultaneously. We present the steps of the accelerating version of the fast SaWL for graph classification in Algorithm 1 of the Appendix. We additionally detail how to use the version for graph isomorphism testing in Appendix A.7.

### 3.3   The Expressive Power of SaWL

We first analyze the expressive power of SaWL by comparing it with 1-WL. Once the graphs can be discriminated by 1-WL, they can be discriminated by SaWL as well.

**Proposition 1.**   *Given two graphs $G$ and $H$, if they can be distinguished by 1-WL, i.e., $\phi^{(h)}(G) \neq \phi^{(h)}(H)$, then they must be distinguished by the SaWL, i.e., $\psi^{(h)}(G) \neq \psi^{(h)}(H)$.*

See Appendix A.2 for proof. If the graph pair can be discriminated by 1-WL, the counting mappings of the whole graphs are different. There must exist subgraphs with different counting mappings in the graph pair. Therefore, the final feature mappings of the two graphs obtained by SaWL are different.

**Proposition 2.** *We denote a multiset for graphs $G$ as $M_G^h$, where the elements are multisets of labels of nodes within $h$-hop subgraphs, i.e., $M_G^h = \{\!\{\{\!\{l_p^h | p \in \mathcal{V}(G_v^h)\}\!\}\}|v \in \mathcal{V}(G)\}\!\}$. For graphs $G$ and $H$, if $M_G^h \neq M_H^h$, then the two graphs can be distinguished by the $h$-layer SaWL.*

In SaWL, $S$ operator encodes a subgraph by hashing the labels of nodes within the subgraph. Then an injective readout function is applied to subgraphs around each node in the graph to obtain the final graph feature mapping. Once any element of multisets $M_G^h$ and $M_H^h$ are different, the two graphs can be distinguished by SaWL. We provide a detailed explanation in Appendix A.3.

**Theorem 1.** *The expressive power of SaWL is higher than that of 1-WL in distinguishing graphs.*

As proved in Proposition 1, once the graphs can be discriminated by 1-WL, they must be discriminated by SaWL. There are also many graphs that can be discriminated by SaWL, but not by 1-WL, e.g., graphs $G$ and $H$ in Figures 2, we provide more examples in Appendix A.4. To sum up, the expressive power of SaWL is strictly higher than that of 1-WL. According to recent research on subgraph GNNs [26], SaWL's $k$-hop subgraph selection and encoding scheme can be implemented by 3-order Invariant Graph Networks (3-IGNs), whose expressive power is bounded by 3-WL [27]. Thus, SaWL's expressive power is also bounded by 3-WL.

### 3.4 Complexity

We analyze the computational complexity of the fast SaWL and the corresponding accelerating version respectively. Given the graph $G$ with node number $N$, average node degree $D$ and edge number $M$, where $M = ND$. We assume the average node number of the subgraphs is $n$. For **the fast SaWL**, the multiset determination, the label compression and relabeling in the WL encoder take a total runtime of $O(ND)$ [18]. In the $S$ operator, the feature mapping computing of one subgraph with $n$ nodes takes $O(n)$, and that of the $N$ subgraphs takes $O(Nn)$. To sum up, the time complexity is $O(ND) + O(Nn)$. For **the accelerating version**, the $S$ operator can be executed simultaneously with the multiset determination of the WL encoder. Specifically, determining the label multisets and identity sets for all nodes takes $O(ND)$ operations which can be accomplished simultaneously. The runtime of the identity set can be achieved by using a hash table. Therefore, the total time complexity of the accelerating version is $O(ND)$, which equals that of 1-WL algorithm [18].

## 4 Subgraph-aware Graph Neural Network

In order to generalize SaWL to scenarios with continuous features, we propose a neural version of SaWL, namely Subgraph-aware GNN (SaGNN). Each component in the SaWL is replaced with a neural network in SaGNN.

**Model.** The neural version SaGNN includes two components: the GNN encoder and the $S$ operator. Any standard neural version of the 1-WL algorithm can be utilized as the GNN encoder. Given input graphs, ***GNN encoder*** updates nodes with its previous state and representations of neighbor nodes (Eq. 1). Specifically, we adopt GIN with $\epsilon = 0$ to obtain the node representations in the $k$-th layer, i.e., $\boldsymbol{h}_v^{(k)} = \mathrm{MLP}^{(k)}\left(\boldsymbol{h}_v^{(k-1)} + \sum_{u \in \mathcal{N}(v)} \boldsymbol{h}_u^{(k-1)}\right)$, where $\mathcal{N}(v)$ denotes the neighbors of node $v$, and $\boldsymbol{h}_v^{(k)} \in \mathbb{R}^{N \times D_1}$, $D_1$ is the feature dimension. In each layer, node representations are updated by the GNN encoder applied to the full graph.

With the updated node representations, $S$ ***operator*** in SaGNN are designed to further encode $k$-hop subgraphs around each node, which provides extra expressive power beyond plain GNN. An injective function is utilized for encoding subgraph information by aggregating nodes within the subgraph (Eq. 3). In this paper, we adopt MLP with SUM as the hash function, as given the input from the countable space, the combination achieves injective [13]. The representation of the subgraph around node $v$ is obtained by $\boldsymbol{h}_{s,v}^{(k)} = \mathrm{MLP}\left(\sum_{q \in \mathcal{V}(G_v^k)} \boldsymbol{h}_q^{(k)}\right)$.

Then, graph representations in the $k$-th layer are calculated with a readout (pooling) function (Eq. 4). In SaGNN, we adopt sum pooling as the readout function, i.e., $\boldsymbol{H}^{(k)}(G) = \mathrm{SUM}\left(\boldsymbol{h}_{s,v}^{(k)} | v \in V(G)\right)$. Then the representations of graph $G$ in all layers are concatenated as the final graph representation, i.e., $\boldsymbol{H}(G) = \mathrm{CONCAT}\left(\boldsymbol{H}^{(1)}(G), \boldsymbol{H}^{(2)}(G), ..., \boldsymbol{H}^{(k)}(G)\right)$, and $\boldsymbol{H}(G) \in \mathbb{R}^{D_1 * k}$.

**Discussion.** Since the SaGNN is the neural version of SaWL, and the SaWL have been shown to be more expressive than 1-WL, the expressive power of SaGNN is higher than that of 1-WL. The computational complexity of SaGNN is also the same as the fast SaWL, which is $O(ND + Nn)$. Besides, both the proposed SaGNN and the existing methods of WL-on-subgraph paradigm [15–17] intend to uplift GNNs by encoding subgraphs. However, methods of WL-on-subgraph paradigm bring high computational cost by extracting rooted subgraphs and applying multiple GNNs. Instead, SaGNN encodes rooted subgraphs with the nodes updated in full graphs, which keeps the computational cost low. We present a detailed comparison in Appendix A.5.

## 5  Experiments

In this section, we first evaluate the effectiveness of the proposed methods on graph classification tasks. Then we conduct experiments to verify the expressiveness and efficiency of our methods.

### 5.1  Datasets

In the tasks of graph classification, we evaluate fast SaWL and SaGNN with seven datasets, including TU datasets [28], and Open Graph Benchmark (OGB) dataset [29]. Graphs in these datasets represent chemical molecules, nodes represent atoms, and edges represent chemical bonds. TU datasets include MUTAG [30], PTC_MR [31], Mutagenicity [32], NCI1 [33] and NCI109 [33]. The task is binary classification, and the metric is classification accuracy. Task on OGB dataset ogbg-molhiv is molecular prediction with metric of ROC-AUC. We evaluate the expressiveness of our methods on the EXP [34], CSL [35] and SR25 datasets [36], which are three synthetic datasets containing 1-WL indistinguishable regular graphs. We provide more description of the datasets in Appendix A.6.

### 5.2  Baselines

In the experiment of the graph classification task on TU, we adopt three graph kernel methods, some GNNs methods based on the 1-WL, and some methods with higher expressive power than 1-WL as baselines. Graph kernel methods include shortest path kernel [37], WL subtree kernel [18] and deep graph kernel [38]. GNNs methods based on the 1-WL include GCN [22], GIN [13], Diffpool [25], and Sortpool [39]. For GCN, graph representations are obtained by the learned nodes representations and sum pooling. Higher expressive methods include 1-2-3 GNN [14], 3-hop GNN [17] Nested GNN [15] and GraphSNN [40]. On OGB dataset, we compare with the traditional message passing GNNs, and the higher expressive methods Deep LRP-1-3 [41], Nested GNN [15] and GIN-AK$^{+}$ [16]. Results of baselines are obtained either from raw paper or source code with published experimental settings ("-" indicates that results are not available). For GCN and GIN, we search the model layer in $\{2, 3, 4, 5\}$, and hidden dimensions in $\{32, 64, 128\}$. For Nested GNN, we choose the best-performing Nested GIN as the baseline according to the results in the original paper. And the results on the datasets Mutagenicity, NCI and NCI109, we search the subgraph height in $\{2, 3, 4, 5\}$ with 4 model layers.

### 5.3  Experimental Setup

For fast SaWL and SaGNN, we adopt multilayer perceptrons (MLPs) with softmax as the classifier to predict the class label of the graph. We also take our SaWL with linear SVM as a graph kernel method named SaWL Kernel to compare with existing graph kernel methods. On the TU datasets, we perform 10-fold cross-validation where 9 folds for training, 1 fold for testing. $10\%$ split of the training set is used for model selection [42]. We report the average and standard deviation (in percentage) of test accuracy across the 10 folds. We train the models with batch size 32. On the OGB dataset, the experiments are conducted 10 times, and the average scores of ROC-AUC are reported. We train the models with batch size 256. For all datasets, we implement experiments with PyTorch and employ Adam optimizer with the learning rate of 0.001 to optimize the model. We search the iteration times of our methods in $\{2, 3, 4\}$. In the training process, we adopt the early stopping strategy with patience 30, and we report the test results at the epoch of best validation. The experimental setups of the expressive power evaluation on the EXP, CSL and SR25 are kept the same with [34–36]. We run all the experiments with Nvidia V100 GPUs.

**Table 1:** 10-Fold Cross Validation average test accuracy (%) on TU datasets.

| Methods | MUTAG | PTC_MR | Mutagenicity | NCI1 | NCI109 |
|---|---|---|---|---|---|
| SP kernel | $87.28 \pm 0.55$ | $58.24 \pm 2.44$ | $71.63 \pm 2.19$ | $73.47 \pm 0.21$ | $73.07 \pm 0.11$ |
| WL kernel | $82.05 \pm 0.36$ | $57.97 \pm 0.49$ | - | $82.19 \pm 0.18$ | $82.46 \pm 0.24$ |
| DGK | $87.44 \pm 2.72$ | $60.08 \pm 2.55$ | - | $73.55 \pm 0.51$ | $73.26 \pm 0.26$ |
| GCN | $78.69 \pm 6.56$ | $66.73 \pm 4.65$ | $80.84 \pm 1.35$ | $78.39 \pm 1.79$ | $77.57 \pm 1.79$ |
| GIN | $81.51 \pm 8.47$ | $54.09 \pm 6.20$ | $77.70 \pm 2.50$ | $80.0 \pm 1.40$ | $70.20 \pm 3.21$ |
| Diffpool | $80.00 \pm 6.98$ | $57.14 \pm 7.11$ | $80.55 \pm 1.98$ | $78.88 \pm 3.05$ | $76.76 \pm 2.38$ |
| SortPool | $85.83 \pm 1.66$ | $58.59 \pm 2.47$ | $80.41 \pm 1.02$ | $74.44 \pm 0.47$ | - |
| 1-2-3-GNN | $86.10 \pm 0.0$ | $60.9 \pm 0.0$ | - | $76.2 \pm 0.0$ | - |
| 3-hop GNN | $87.56 \pm 0.72$ | - | - | $80.61 \pm 0.34$ | - |
| Nested GIN | $87.90 \pm 8.20$ | $54.1 \pm 7.70$ | $82.40 \pm 2.00$ | $78.60 \pm 2.30$ | $77.20 \pm 2.90$ |
| GraphSNN | $\mathbf{91.57 \pm 2.80}$ | $66.70 \pm 3.70$ | - | $81.60 \pm 2.80$ | - |
| **SaWL Kernel** | $87.31 \pm 7.04$ | $63.40 \pm 7.30$ | $81.05 \pm 1.96$ | $\mathbf{83.80 \pm 1.80}$ | $82.48 \pm 2.54$ |
| **fast SaWL** | $\mathbf{90.00 \pm 3.89}$ | $\mathbf{70.33 \pm 5.32}$ | $\mathbf{84.32 \pm 1.48}$ | $\mathbf{84.45 \pm 0.66}$ | $\mathbf{85.37 \pm 0.81}$ |
| **SaGNN** | $88.81 \pm 5.21$ | $\mathbf{71.78 \pm 4.43}$ | $\mathbf{84.13 \pm 1.31}$ | $83.78 \pm 1.03$ | $\mathbf{83.35 \pm 0.56}$ |

**Table 2:** Performance Evaluation on OGB dataset.

| Methods | ogbg-molhiv (AUC) | |
|---|---|---|
| | Validation | Test |
| GCN [22] | $82.04 \pm 1.41$ | $76.06 \pm 0.97$ |
| GIN [13] | $82.32 \pm 0.90$ | $75.58 \pm 1.40$ |
| Deep LRP-1-3 [41] | $81.31 \pm 0.88$ | $76.87 \pm 1.80$ |
| Nested GNN [15] | $83.17 \pm 1.99$ | $78.34 \pm 1.86$ |
| GIN-AK$^+$ [16] | - | $\mathbf{79.61 \pm 1.19}$ |
| **fast SaWL** | $79.13 \pm 0.69$ | $78.29 \pm 0.48$ |
| **SaGNN** | $81.06 \pm 1.14$ | $78.86 \pm 0.73$ |

**Table 3:** Evaluation of Expressiveness.

| Methods | EXP (ACC) | CSL (ACC) | SR25 (ACC) |
|---|---|---|---|
| GCN [22] | $50.0 \pm 0.00$ | $10.0 \pm 0.00$ | $6.67$ |
| GIN [13] | $50.0 \pm 0.00$ | $10.0 \pm 0.00$ | $6.67$ |
| GCN-RNI [34] | $98.0 \pm 1.85$ | $16.0 \pm 0.00$ | $6.67$ |
| PPGN [43] | $\mathbf{100.0 \pm 0.00}$ | - | $6.67$ |
| 3-GCN [14] | $99.7 \pm 0.004$ | $95.70 \pm 14.85$ | $6.67$ |
| Nested GNN [15] | $99.9 \pm 0.26$ | - | $6.67$ |
| GIN-AK$^+$ [16] | $\mathbf{100.0 \pm 0.00}$ | - | $6.67$ |
| **fast SaWL** | $99.50 \pm 0.70$ | $80.67 \pm 8.04$ | $6.67$ |
| **SaGNN** | $99.67 \pm 0.70$ | $84.67 \pm 10.45$ | $6.67$ |

## 5.4 Effectiveness Evaluation

**Performance on Graph Classification Task.** Results of the graph classification on TU and OGB datasets are shown in Tables 1, 2. On TU datasets, our SaWL kernel gains strong improvements compared with graph kernel methods and traditional GNNs based on 1-WL. Especially, SaWL kernel achieves better performance than WL subtree kernel on all TU datasets and it outperforms WL subtree kernel by more than $10\%$ in accuracy on Mutagenicity, NCI1 and NCI109, which proves the effectiveness of the $S$ operator experimentally. It verifies that the augmented subgraph information on the basis of the subtree pattern enhances the expressive power of our model in the graph classification task. For methods with higher expressive power than traditional message passing GNNs, i.e., 1-2-3-GNN, 3-hop GNN, Nested GNN and GraphSNN, our fast SaWL and SaGNN still outperform the methods on most TU datasets. Especially, our fast SaWL gains such progress with low computational cost. On the larger-scale OGB dataset, our methods achieve comparable performance at a lower cost compared with other highly expressive methods. We adopt GIN as the GNN encoder in SaGNN. And the improvements compared to GIN verify the effectiveness of the $S$ operator, which provides additional subgraph information and improves the distinguishing power in graph classification. The neural version SaGNN achieves slightly lower performance than fast SaWL on some small-scale datasets, which may be because the neural model is not sufficiently trained with insufficient training data. On the larger-scale OGB dataset, the neural version SaGNN achieves better results than fast SaWL with sufficient training. In summary, the proposed fast SaWL and SaGNN achieve improvement compared with competitive baselines in the graph classification task.

**Expressive Power Evaluation.** We first evaluate the expressiveness on the EXP, CSL and SR25 datasets, and then show some cases of graph isomorphism testing in Appendix A.7. Results of empirical evaluation are shown in Table 3, and some results of baselines are from [34, 44]. Each pair of graphs in the three datasets is non-isomorphic and 1-WL indistinguishable, and the results of GCN and GIN verify this. We adopt five methods with high expressive power as baselines [14–16, 34, 43]. On EXP, our fast SaWL and SaGNN consistently achieve very high accuracy, which can distinguish nearly all graph pairs. The results are comparable with the $k$-GNNs [14, 43] and Nested GNN [15], which are more computationally complex. On CSL, our methods significantly outperform 1-WL based GNNs and are lower than 3-GCN. The results verify the high distinguishing power of fast

SaWL and SaGNN, which have been stated theoretically. Strongly regular graphs in SR25 are 3-WL equivalent [45] and cannot be distinguished by the methods in Table 3.

## 5.5 Efficiency Evaluation

We compare the running time of the proposed methods with baselines to verify the high efficiency in practice. Our **fast SaWL** has higher discriminating power than 1-WL, while the accelerating version of the fast SaWL has the same time complexity as 1-WL, which have been demonstrated in section 3.4. We record the running time of fast SaWL and 1-WL in obtaining feature mappings of all graphs in four datasets respectively. The average running time with ten runs are shown in Tabel 4. The running time of fast SaWL is similar to that of 1-WL. The time difference is less than $0.5$ seconds on the TU dataset and less than 3 seconds on the ogbg-molhiv, which contains 41127 graphs.

We further conduct the t-test as a significance test. The p-value is $0.8413$, and $0.8413 > 0.05$. The experimental results demonstrate that there is no significant difference in the running time of fast SaWL and 1-WL on graph feature mapping computation, which is consistent with the theoretical analysis.

**Table 4:** Runtime Comparison of fast SaWL with 1-WL on Graph Feature Mapping Computation (second).

| Methods | Mutagenicity | NCI1 | NCI109 | ogbg-molhiv |
|---------|--------------|------|--------|-------------|
| 1-WL | $4.90 \pm 0.23$ | $4.69 \pm 0.16$ | $4.73 \pm 0.20$ | $112.25 \pm 0.68$ |
| **fast SaWL** | $4.99 \pm 0.22$ | $4.81 \pm 0.20$ | $4.96 \pm 0.20$ | $115.11 \pm 0.71$ |

For **SaGNN**, we compare the running time with an example method of the WL-on-subgraph paradigm, i.e., Nested GNN (NGNN) [15]. Results are shown in Figure 4. For a fair comparison, we set the model layer and hidden dimension the same. On TU datasets, the running time of the Nested GNN is more than three times that of SaGNN. On the ogbg-molhiv dataset (abbreviated as ogb in Figure 4), we compare the epoch time and the whole training time. The running time of the Nested GNN is more than eight times that of SaGNN on both each epoch and the whole training process, e.g., the average training time of Nested GNN on an epoch is

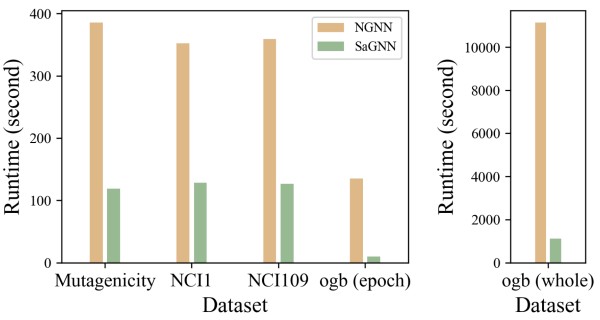

**Figure 4:** Training Time Comparison of SaGNN with Method of the WL-on-subgraph paradigm.

134.91 seconds, and that of SaGNN is 9.71 seconds, the whole training time of Nested GNN is 10331 seconds and that of SaGNN is 1206 seconds. We also test GNN-AK without subgraph sampling modules on ogbg-molhiv. The whole training time is 6100 seconds, which is five times that of SaGNN. The time comparison empirically demonstrates that our SaGNN is significantly more efficient than methods of the WL-on-subgraph paradigm and has a better generalization to large-scale graphs.

## 6 Related Works

Graph classification is an important task with many valuable downstream applications, such as chemical molecular property prediction [46] and pharmaceutical drug research [2]. Graph classification aims to predict the labels of given graphs by utilizing graph structure and feature information.

**Graph kernels.** Historically, graph kernels have been the dominant approaches for graph classification. Graph kernels first decompose the graph into different substructures, e.g., path, graphlet, and subtree. Then kernel matrix of graphs is calculated by implicit way with kernel functions or explicit way with graph feature mappings. Finally, kernel matrix is sent to kernel machine to obtain the predicted labels of graphs. Typical graph kernel methods include shortest path kernel [37], random walk graph kernel [47], graphlet kernel [48], and WL subtree kernel [18]. However, graph kernel methods have limitations in graph classification due to heuristic feature extraction.

**GNNs based on 1-WL algorithm.** Recently, Graph Neural Networks (GNNs) have been popular methods for graph classification, which made a great success [39, 49]. These methods can be viewed as the neural implementation of the 1-WL [13, 22], which first updates node representations by aggregating neighbor nodes, and then pools the nodes to obtain the graph representation. Many pooling strategies have been proposed for graph classification [23, 25, 50]. However, it has been proved that the expressive power of traditional GNNs based on 1-WL is at most as large as 1-WL [13, 14], which limits the performance of GNN-pooling methods on the graph classification task.

**High Expressive GNNs.** The expressiveness of GNNs is a key research topic in graph machine learning. Many approaches with higher expressive power than 1-WL have been proposed, including high-dimension WL based [14, 43], feature augmentation based [34, 51], subgraph encoding based [15, 16, 52] and some equivariant models [26, 27, 53]. We provide a brief review here. **(1)** It's natural to build GNNs based on a high-dimensional WL algorithm for high expressive power, e.g., PPNG [43] based on the high-order graph networks, $k$-GNNs [14] based on the set $k$-WL algorithm. However, high-dimensional WL algorithms require enumeration of the node tuples, which limits the scalability and generalization with high computational cost. **(2)** Some researchers propose to improve the expressive power of GNN by adding additional features. They augment GNNs by concatenating pre-extracted sub-structural information or random features as additional node features [34, 41, 54]. For example, Graph Structure Networks (GSN) [54] encodes structural information in the additional preprocessing stage by counting the appearance of certain substructures as the structural feature vector. Then the structural features are utilized in message passing. GCN-RNI [34] enhances GNNs with random node initialization. rGINs [51] concatenates random features with node features and then applies GINs on the combined features. However, such additional feature augmentation-based methods limit the generalization ability of the methods. **(3)** Many existing subgraph-based methods first extract subgraphs centered on each node of graphs, then apply GNNs on the extracted subgraphs [15, 16]. E.g., Nested GNN [15] implements base GNN on the extracted subgraphs then obtains the whole graph representations by a global pooling. These methods can be summarized as WL-on-subgraph paradigm (Figure 1 (b)), and the computational cost are much higher than 1-WL, which limits their application to the large scale graphs. We provide more related works in Appendix A.9.

## 7   Conclusion

The traditional message passing graph neural networks (GNNs) are at most as powerful as 1-WL test. Since the representative power of the subgraph is higher than that of the subtree, methods of the WL-on-subgraph paradigm are proposed to improve GNNs, which brings expensive computational cost. As a contrast, we propose a subgraph-aware WL (SaWL) paradigm in this paper, which uplifts GNNs and keeps computation complexity low. Under the paradigm, we first implement an algorithm named fast SaWL, where the additional $S$ operator encodes subgraph information on the basis of the WL on the full graph. We then present the neural version of the SaWL named SaGNN, which replace the components in SaWL with neural networks. SaWL and SaGNN are proved to be more expressive than 1-WL, and have achieved significant improvements in the experiments.

## Acknowledgements

The authors would like to thank the anonymous reviewers for their valuable suggestions. This work is funded by the National Natural Science Foundation of China under Grant Nos. U21B2046, 62272125, and the National Key R&D Program of China (2020AAA0105200). Huawei Shen is also supported by Beijing Academy of Artificial Intelligence (BAAI).

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

# A Appendix

## A.1 Acceleration of the fast SaWL

The $S$ operator in the fast SaWL (Section 3.2) can be calculated simultaneously with the WL encoder, which leads to the accelerating version. The idea of the acceleration is illustrated in Figure 3, each node contributes to the feature mappings of $m$ rooted subgraphs, where $m$ equals the size of rooted subgraph centered in the node. The sizes of rooted subgraphs can be computed simultaneously with the multiset determination of the WL encoder. We then present the steps of the accelerating version.

The accelerating version proceeds in iterations. Each iteration consists of **five steps** (Algorithm 1), which are multisets determination, multisets sorting, label compression, relabeling and feature mapping obtaining. Specifically, given two graphs $G$ and $G'$, for node $v$, the label is denoted as $l_v^h$ and the identity is denoted as $id_v$. In **step 1**, we aggregate the labels and identity sets of neighbor nodes respectively. Node labels of neighbor nodes are aggregated as a multiset $M_v^h$. For $h = 0$, $M_v^0 = l_v^0$, and for $h > 0$, $M_v^h = \{\!\{l_u^{h-1}|u \in \mathcal{N}(v)\}\!\}$, where $\mathcal{N}(v)$ denotes the neighbor nodes of $v$ and $\{\!\{\}\!\}$ denotes a multiset. Identity sets of neighbor nodes are aggregated and combined with the identity of the center node which forms a new set $t_v^h$. For $h = 0$, $t_v^0 = \{id_v\}$, and for $h > 0$, $t_v^h = \{id_v, id_w|w \in t_u^{h-1}, u \in \mathcal{N}(v)\}$. In **step 2**, each label multiset $M_v^h$ is sorted and converted to a string $\mathcal{S}_v^h$ with the prefix $l_v^{h-1}$, which prepares for the label compression. In **step 3**, each string is compressed to a new label with a hash function $g : \sum * \to \sum$ and $g$ should be an injective function. The mapping alphabet is shared across graphs, which guarantees a common feature space. In **step 4**, we relabel each node in graph $G$ and $G'$ as $l_v^h := g(\mathcal{S}_v^h)$.

We assume the minimum label in $h$-th iteration is $l_m$. Then, in **step 5**, we compute the graph feature mapping. The value of the $i$-th position ($i$ starts from 0) of the feature mapping in layer $h$ is:

$$\psi_i^{(h)}(G) = \sum_{l_v^h = l_m + i, v \in V} \left| t_v^h \right|, \tag{5}$$

which means the summation of the occurrences of label $l_m + i$ in all $h$-hop subgraphs. The final graph feature mappings obtained by the fast SaWL and the accelerating version are equivalent. In the accelerating version, the feature mappings of subgraphs do not require to be calculated separately, which reduces the computational cost and speeds up the computation.

---

**Algorithm 1** Accelerating version of fast SaWL for Graph Classification

---

**Input:** Node Labels (features) $\mathbf{X}$; Adjacency Matrix $\mathbf{A}$

  **for** $h = 1$ **to** $H$ **do**

    1. Label multisets and identity sets determination

- Aggeregate labels of neighbor nodes centered in each node $v$ in graph $G$ as multiset $M_v^h$. For $h = 0$, $M_v^0 = l_v^0$, and for $h > 0$, $M_v^h = \{\!\{l_u^{h-1}|u \in \mathcal{N}(v)\}\!\}$.
- Aggregate identity sets of neighbor nodes centered in each node $v$ in graph $G$. Identity of node $v$ and elements in identity sets of neighbor nodes compose the new identity set. For $h = 0$, $t_v^h = \{id(v)\}$, for $h > 0$, $t_v^h = \{id_v, id_w|w \in t_u^{h-1}, u \in \mathcal{N}(v)\}$.

    2. Sorting labels in each label multiset

- Sort label elements in the label multiset in ascending order and concatenate them into a string $\mathcal{S}_v^h$.
- Add $l_v^{h-1}$ as a prefix to $\mathcal{S}_v^h$.

    3. Label compression

- Map each string $\mathcal{S}_v^h$ to a compressed label using a hash function $g : \sum * \to \sum$ such that $g(\mathcal{S}_v^h) := g(\mathcal{S}_w^h)$ if and only if $\mathcal{S}_v^h = \mathcal{S}_w^h$.

    4. Relabeling

- Set $l_v^h := g(\mathcal{S}_v^h)$ for all nodes in the graph.

    5. $i$-th position of graph feature mapping of $h$ layer

- $\psi_i^{(h)}(G) = \sum_{l_v^h = l_m + i, v \in V} \left| t_v^h \right|$.

  **end for**

**Output:** Graph Feature Vector $\psi(G) = \left[ \psi^{(0)}(G), ..., \psi^{(H)}(G) \right]$

---

## A.2 Proof of Proposition 1

*Proof.* For graphs $G$ and $H$, if they can be discriminated by 1-WL, there must exits a constant $h$ that $\phi^{(h)}(G) \neq \phi^{(h)}(H)$. Since $\phi^{(h)}(G) = (c_h(G, \ell_1^h), ..., c_h(G, \ell_{|\mathcal{L}_h|}^h))$, there must exist a $\ell_i^h$, such that $c_h(G, \ell_i^h) \neq c_h(H, \ell_i^h)$. Then there must be different subgraphs in the two graphs such that $c_h(G_v^h, \ell_i^h) \neq c_h(H_u^h, \ell_i^h)$, where $G_v^h$ is a $h$-hop subgraph around node $v$ of $G$. As a result, the sets of subgraph feature mappings of graph $G$ and $H$ are not equal, i.e., $\{\phi(G_v^h)|v \in \mathcal{V}(G)\} \neq \{\phi(H_u^h)|u \in \mathcal{V}(H)\}$. With the condition that READOUT is a injective function, we have READOUT($\{\phi(G_v^h)|v \in V(G)\}$) $\neq$ READOUT($\{\phi(H_u^h)|u \in \mathcal{V}(H)\}$), i.e., $\psi_h(G) \neq \psi_h(H)$. In other words, the graph $G$ and $H$ can also be discriminated by the SaWL.

## A.3 Explaination of Proposition 2

We further explain the Proposition 2 in section 3.3. For graphs $G$ and $H$, if $\{\!\{\{\!\{l_p^h|p \in \mathcal{V}(G_v^h)\}\!\}|v \in \mathcal{V}(G)\}\!\} \neq \{\!\{\{\!\{l_q^h|q \in \mathcal{V}(G_u^h)\}\!\}|u \in \mathcal{V}(H)\}\!\}$, then the two graphs can be distinguished by the $h$-layer SaWL. Inner multisets $\{\!\{l_p^h|p \in \mathcal{V}(G_v^h)\}\!\}$ and $\{\!\{l_q^h|q \in \mathcal{V}(G_u^h)\}\!\}$ encode the subgraphs information of graph $G$ and $H$ by $S$ operator. Once any pair of the encoded subgraphs differs, readout function can map the two graphs to different feature mappings. The graphs $G$ and $H$ can be distinguished by SaWL.

We provide a simplified and specific version of Proposition 2. We assume the inner multisets encode the subgraph information only with the number of nodes within the subgraph. We define the number of $h$-shortest neighbors of each node as $s_v^h$, which is the number of nodes with the exact shortest distance $h$ from the center node $v$. For graphs $G$ and $H$, if $\{\!\{s_v^h|v \in \mathcal{V}(G)\}\!\} \neq \{\!\{s_u^h|u \in \mathcal{V}(H)\}\!\}$, then the two graphs can be distinguished by the $h$-layer fast SaWL.

For graphs $G$ and $H$, if $\{\!\{s_v^h|v \in \mathcal{V}(G)\}\!\} \neq \{\!\{s_u^h|u \in \mathcal{V}(H)\}\!\}$, then the two graphs can be distinguished by the $h$-layer SaWL. $s_v^h$ is the number of nodes with the exact shortest distance $h$ from node $v$. When $h = 1$, if the numbers of 1-hop neighbor nodes are different in $G$ and $H$, 1-WL can discriminate the two graphs, i.e., $\phi^{(h)}(G) \neq \phi^{(h)}(H)$. According to Proposition 1, SaWL can discriminate the graphs as well. Assume the numbers of 1-hop neighbor nodes are the same, when $h = 2$, the number of nodes with the shortest distance 2 are different in $G$ and $H$. Then the sizes of 2-hop rooted subgraphs in $G$ and $H$ are different, which leads to the difference in the multisets of rooted subgraphs in the two graphs. With the injective readout function, the final graph feature mappings of the graph $G$ and $H$ are different. Similarly, assume the numbers of $(h-1)$-hop neighbor nodes in two graphs are the same. Then if the numbers of $h$-shortest distance nodes in two graphs are different, it results in the different multisets of rooted subgraphs and the different graph feature mappings. Therefore, the graphs $G$ and $H$ can be discriminated by SaWL.

For a further intuitive understanding, we take the implemented algorithm of SaWL, i.e., fast SaWL, as an example. From the perspective of the accelerating version, the size of the rooted subgraph equals the contribution of the center nodes to the whole graph feature mapping (shown in Figure 3(b)). Therefore, different sizes of rooted subgraphs directly lead to different feature mappings of the graph $G$ and $H$. The graphs can be discriminated by fast SaWL.

## A.4 Graph Examples

In this subsection, we provide two classes of graphs that cannot be discriminated by WL [12], but can be discriminated by the proposed SaWL. Note that the labels of all nodes in Figure 5 are the same. SaWL can discriminate the graphs only by utilizing the graph structure, and the additional label information of nodes can leave the discrimination easier.

The first class is $k$-regular graphs of the same size (Figure 5(b)-(f)). The 6-nodes 2-regular graph in Figure 5(b), 8-nodes 3-regular graphs in Figure 5(c), 12-nodes 4-regular graphs in Figure 5(d) and two pairs of circulant graphs in Figure 5(e), 5(f) can be discriminated by 2-layer SaWL. The green nodes are center nodes, and the grey nodes are 2-hop neighbors of the green nodes. We take Figure 5(c) as example. There are two 2-hop shortest neighbors of the green node in the left graph, which are marked as grey. While for the green node in the right graph, the number of the 2-hop shortest neighbor is three (grey nodes in the right graph). According to proposition 2 in section 3.3, the left graph and the right graph can be discriminated by SaWL with two layers.

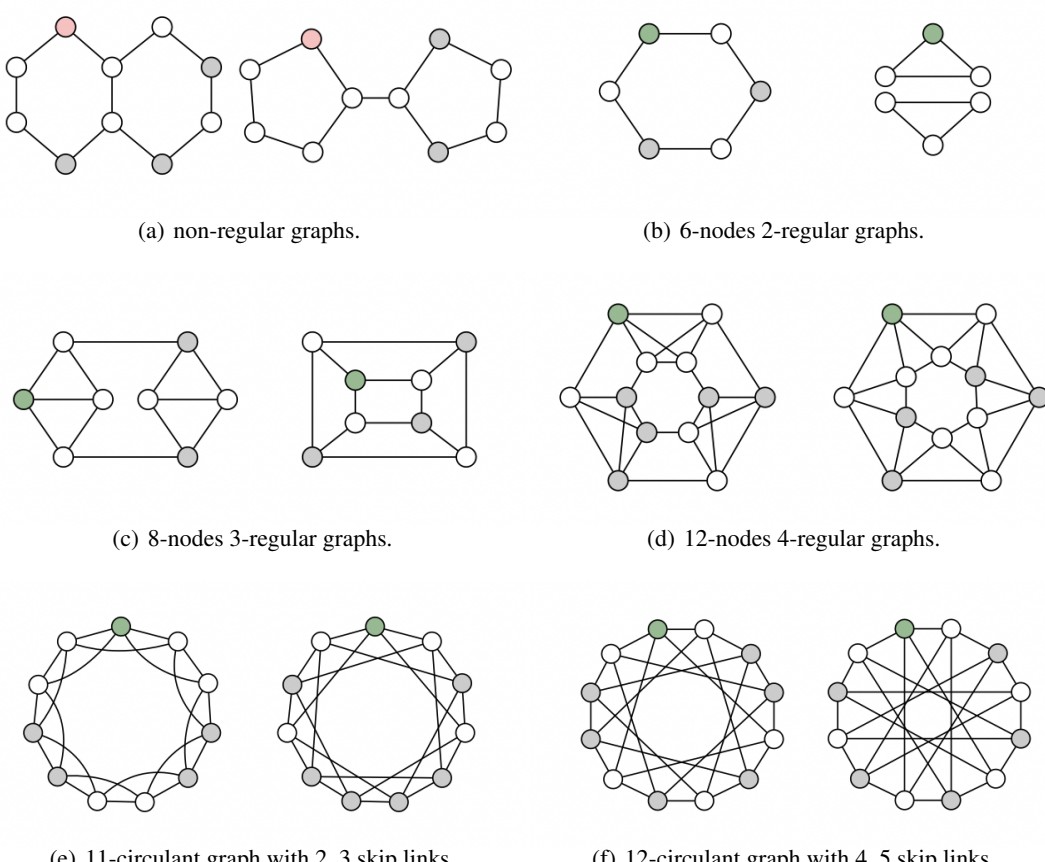

(a) non-regular graphs.

(b) 6-nodes 2-regular graphs.

(c) 8-nodes 3-regular graphs.

(d) 12-nodes 4-regular graphs.

(e) 11-circulant graph with 2, 3 skip links.

(f) 12-circulant graph with 4, 5 skip links.

**Figure 5:** Graph pairs can discriminated by SaWL, but not WL.

For a more intuitive understanding, we present the feature mappings of graphs in Figure 5(c) with 1-WL and our fast SaWL. We assume the initial label of each node is $0$. For 1-WL, the multiset determination in the 1st and the 2nd iteration includes $0, 000 \rightarrow 1; 1, 111 \rightarrow 2$. The feature mappings of the graph in the left and right after the 2nd iteration are equal, i.e., $\phi(G_{left}) = \phi(G_{right}) = (8, 8, 8)$. For our fast SaWL, the feature mapping of the graph in the left is $\psi(G_{left}) = (8, 32, 52)$, while that of the right graph is $\psi(G_{right}) = (8, 32, 56)$. The difference comes from the green node and its equivalent nodes. In the left graph, label $2$ occurs $52$ times in all rooted subgraphs, and it occurs $56$ times in the rooted subgraphs of the right graph.

The second class includes some non-regular non-isomorphic graphs, e.g., Figure 5(a). The two graphs are non-regular graphs, but WL cannot distinguish them. SaWL can discriminate the two graphs with three layers. We take pink nodes as center nodes. For the left graph, there are three 3-hop shortest neighbors of the pink node. While for the right graph, there exist two 3-hop shortest neighbors of the pink node, which are marked as grey. Therefore, the two graphs can be distinguished by SaWL.

### A.5 Comparison with WL-on-subgraphs methods

We discuss relations of the proposed methods of subgraph-aware WL (Figure 1(c)) paradigm with other methods of WL-on-subgraph paradigm (Figure 1(b)). Methods of WL-on-subgraph paradigm usually extract subgraphs around each node of the graph, then apply GNNs on each extracted subgraph respectively, such as Nested GNN [15], GNN-AK [16] and k-hop GNN [17]. However, the computation complexity of this paradigm is much higher than our proposed subgraph-aware WL paradigm. Given a graph $G$ with $N$ nodes, the average degree of nodes is denoted as $D$, and the average nodes number of subgraphs is denoted as $n$. Extracting $k$-hop subgraphs from each node

takes $O(k \cdot N \cdot D)$. Applying GNNs on all extracted subgraphs takes $O(N \cdot n \cdot D)$. Totally, the computation cost is $O(k \cdot N \cdot D + N \cdot n \cdot D)$. Compared to high dimensional GNNs based on $k$-WL, methods of WL-on-Subgraph paradigm reduce the computational cost. However, the complexity is still much higher than that of 1-WL and our proposed methods. Essentially, Both the proposed methods of subgraph-aware WL paradigm and the existing methods of WL-on-subgraph paradigm intend to uplift GNNs by encoding subgraphs. However, the WL-on-subgraph methods apply GNNs on all extracted subgraphs respectively, which brings high computational cost. As a contrast, our subgraph-aware WL methods encode subgraphs while keeping the computational cost low.

### A.6  Datasets Description

We provide statistics of the datasets utilized in graph classification tasks in table 5. We adopt molecular datasets for evaluation, including TU datasets and OGB dataset. Nodes in these datasets denote atoms, and the edges denote chemical bonds. To empirically evaluate the expressive power, we adopted EXP [34], CSL [35] and SR25 datasets [36]. **EXP** contains 600 pair of non-isomorphic graphs, which cannot be distinguished by 1-WL. The task is to classify the graphs to 2 classes. **CSL dataset** [35] contains 150 4-regular graphs which cannot be distinguised by 1-WL. Each graph contains 41 nodes with same degree 4 and 164 edges. The task is to classify the regular graphs to 10 isomorphism classes. **SR25 dataset** [36] contains 15 strongly regular graphs. Each graph contains 25 nodes and 300 edges. The task is to classify the regular graphs to 15 different isomorphism classes. There's no node feature and edge feature in these three datasets. The model needs to utilize purely structural information to distinguish graphs.

**Table 5:** Statistics of datasets.

| Dataset | #Graphs | #Positive | #Avg. Nodes | #Avg. Edges | #Nodes Types |
|---|---|---|---|---|---|
| MUTAG | 188 | 125 | 17.9 | 19.8 | 7 |
| PTC_MR | 344 | 152 | 25.6 | 29.4 | 18 |
| Mutagenicity | 4337 | 2401 | 30.3 | 30.8 | 13 |
| NCI1 | 4110 | 2057 | 29.9 | 32.3 | 37 |
| NCI109 | 4127 | 2079 | 29.6 | 32.1 | 38 |
| ogbg-molhiv | 41127 | 1443 | 25.5 | 27.5 | 119 |

### A.7  The Accelerating Version for graph isomorphism testing

The accelerating version of fast SaWL provided in Appendix A.1 can be utilized for the graph isomorphism testing, which has the same time complexity as 1-WL, but higher discriminating power than 1-WL. We first present the definition of the graph isomorphism testing, and then we explain the steps and the termination condition of the accelerating version in the graph isomorphism testing.

**Graph Isomorphism Testing.** Given a graph $G$, $\mathcal{V}(G)$ and $\mathcal{E}(G)$ are the sets of nodes and edges respectively. Two graphs $G$ and $H$ are isomorphic if there exists a bijection $\xi$ between $\mathcal{V}(G)$ and $\mathcal{V}(H)$. $\xi : \mathcal{V}(G) \to \mathcal{V}(H)$ and it preserves the edge relation, i.e., $(u,v) \in \mathcal{E}(G)$ if and only if $(\xi(u), \xi(v)) \in \mathcal{E}(H)$ for all $u, v \in \mathcal{V}(G)$. Although the exact complexity of the graph isomorphism problem is still uncertain, there are some efficient graph isomorphism algorithms [11].

**The Accelerating Version of Fast SaWL for Graph Isomorphism Testing.** When used for the graph isomorphism testing, each iteration of the accelerating version consists of four steps, i.e., steps 1-4 of Algorithm 1. Given graphs $G$ and $H$, the accelerating version terminates after iteration $h$ if:

$$\{(l_v^h, |t_v^h|) | v \in \mathcal{V}(G)\} \neq \{(l_u^h, |t_u^h|) | u \in \mathcal{V}(H)\}. \tag{6}$$

$l_v^h$ denotes the label of node $v$ in the $h$-th iteration, and it represents a $h$-height subtree pattern. $t_v^h$ denotes the set of the node identities. It contains node identities in the subtree pattern without repetition due to the uniqueness of the node identity. The termination condition implies that fast SaWL can determine that two graphs are non-isomorphic once the updated labels or the number of nodes in the subtree patterns are different. The terminating condition of the 1-WL can be denoted

as $\{l_v^h | v \in \mathcal{V}(G)\} \neq \{l_u^h | u \in \mathcal{V}(H)\}$ [18]. The terminating condition of the accelerating version of fast SaWL (Eq. 6) is stricter than that of 1-WL by adding a new structural constraint. Therefore, once the graphs are determined unequal by the 1-WL algorithm, they must also be determined unequal by the proposed implementation. Besides, there exist many graphs that WL cannot discriminate, which can be determined as non-isomorphic (e.g., graph pairs in Figure 5). To conclude, the discriminating power of the SaWL is higher than that of 1-WL in the graph isomorphism testing.

**Cases.** We take the graph pair in Figure 5(c) as an example, the iteration process has been described in Appendix A.4. We denote the left graph as $G$ and the right graph as $H$. After the 2nd iteration, for our fast SaWL, the set of graph $G$, i.e., $\{(2,6),(2,7)|v \in \mathcal{V}(G)\}$ is not equal to the set of graph $H$, i.e., $\{(2,7)|u \in \mathcal{V}(H)\}$. The terminating condition is satisfied, and the two graphs are determined as non-isomorphic. While for 1-WL, $\{2|v \in \mathcal{V}(G)\} = \{2|u \in \mathcal{V}(H)\}$, the two graphs cannot be discriminated. All graph pairs in Figure 5 can be discriminated by fast SaWL in this way.

## A.8 Ablation Study

In this section, we conduct ablation studies on number of iteration. We adopt one dataset for expressive power evaluation and one dataset for graph classification, i.e., CSL [35] and Mutagenicity [32]. We test the performance of our fast SaWL and SaGNN with different numbers of iteration from 1 to 5. We report average accuracy of ten times running in Table 6. $I$=2 denotes two times iterations. From the results, it can be observed that as the number of iterations increases, the performance first improves and then drops a little. It is basically similar on both datasets. The expressive power of the models increases first and then tends to remain unchanged. The methods achieve the best results when the number of iterations is 3 or 4. When the number of iterations is 5, the performance is slightly worse, which may be caused by the increase of the dimension of the feature mapping and the increase of the model parameters. Relatively, the neural version SaGNN requires more iterations than the fast SaWL to get the best results. When the training data is sufficient, SaGNN can achieve better performance, which can be observed in Table 2 as well.

**Table 6:** Ablation Study on Number of Iteration (ACC).

| Datasets | Iteration | $I$=1 | $I$=2 | $I$=3 | $I$=4 | $I$=5 |
|---|---|---|---|---|---|---|
| CSL | fast SaWL | 14.67 | 45.33 | 82.67 | 80.67 | 81.33 |
|  | SaGNN | 12.67 | 23.33 | 56.67 | 84.67 | 80.00 |
| Mutagenicity | fast SaWL | 79.81 | 81.77 | 83.41 | 84.16 | 82.16 |
|  | SaGNN | 79.07 | 81.94 | 83.12 | 84.13 | 83.08 |

## A.9 More Related Works

We present more related works, including substructure encoding based methods and more highly expressive GNNs here. **Substructure encoding based methods.** Some methods utilize subgraph/substructure information as additional node features [41, 54]. For example, Graph Structure Networks (GSN) proposed in [54] encodes structural information in the additional preprocessing stage by counting the appearance of certain substructures as the structural feature vector. Then the structural features are utilized in message passing. The structure encoding in these method is more like a heuristic feature engineering. The selection of the certain substructures requires domain knowledge. This kind of method lacks flexibility and cannot guarantee generalization. It also requires high computational cost as choosing good substructures remains an open problem due to its combinatorial nature. **More highly expressive GNNs.** ESAN [53] encodes a graph by a bag of subgraphs to achieve higher expressive power, which shares some similarities with us. However, ESAN needs some predefined policy to obtain subgraphs. The obtained subgraphs are then encoded by an equivariant architecture. It relies on the subgraph selection policy to achieve high expressivity, which loses some generalization. $K$-hop GNNs [17, 55] propose to aggregate the node with the information from its $k$-hop neighborhood, rather than only from its direct neighbors, which can identity fundamental graph properties such as connectivity and triangle freeness. $K$-hop GNNs leverage multi-hop information to improve the expressive power of GNNs, while it has some differences from methods of WL-on-subgraph. More comparison and discussion can be found in [55].

