# OpenReview forum: "Towards Efficient and Expressive GNNs for Graph Classification via Subgraph-aware Weisfeiler-Lehman"
_logconference.io/LOG/2022/Conference — LoG 2022 Poster_

### Official Review · Reviewer_yoxW · 2022-10-13

**Overall Score:** 6
**Confidence:** 3

**Review:**

****Recommendation to Accept****
########################################################################

Summary:

The paper proposes a subgraph-aware WL scheme by incorporating a subgraph encoding operator after each WL iteration. This new paradigm is shown to be strictly more expressive than 1-WL whilst being computationally faster than approaches based on WL on subgraphs. The authors also propose a neural version of this algorithm which performs competitively on some benchmarks.

########################################################################

Reasons for Score:

The paper does well in describing the merits of the model in terms of computational efficiency and the model performs well on the given benchmarks. However, a better description of how this model fits with current literature is needed.  The runtime and expressivity of its own algorithm is well described but more can be done for expressivity comparisons to existing work on WL on subgraphs and also comparisons to other models such as those using subgraph-encoding schemes.

########################################################################

Strengths

- The paper provides a novel framework for incorporating subgraph information that is computationally efficient whilst being more expressive than 1-WL. The authors clearly show how their model is more expressive than 1-WL and state the complexity of their algorithm and how it compares to WL on subgraphs.
- The model performs very well on the benchmarks given, outperforming the WL-on subgraph methods whilst being faster.

########################################################################

Weaknesses:

- There are other methods that take into account subgraph information and perform a single GNN on the input graph. For example, a subgraph-encoding method such as [1]. What are the benefits of your proposed approach over these subgraph-encoding based models? This would be beneficial to describe in the related work section particularly when you are proposing your model based on being efficient and including subgraph information.
- In the expressivity experiment, both your model and WL on subgraph models achieve around 100%. I don’t think you can claim from this that you have similar expressive power, but more that the task is too easy for all these models. Given that the main point of the paper seems to be comparing to WL on subgraphs with better computational efficiency whilst retaining expressivity; I think a better description of how these models lie between 1-WL and 3-WL and what you are losing (if anything) with a faster method would be beneficial.

########################################################################

Minor Suggestions/Thoughts

- I would say more than one WL on subgraph method should be compared against in terms of computational efficiency. Otherwise it may just look like you have picked the slowest one. For instance in the 3-hop GNN paper they say in practice their model speed is not prohibitive [2]. Either this or state datasets which these models would be too slow on. I just want a clearer understanding of the limitations of these approaches in practice.
- Line 10/Line 11 - I think you can be clearer here by saying beyond 1-WL and quantify your runtime rather than just saying “a fast implementation”
- In your approach you store the intermediate representations of the graph at each layer. If you store the graph representations after each layer of a standard GNN (make subgraph the whole graph) would your approach perform much better and why? Maybe would be nice to have a comparison to this and to have a general understanding of what happens when you change the subgraph size h.

########################################################################

Writing/Spelling

- Line 220: “which achieves injective for the countable feature space” doesn’t make sense to me
- Line 335:  Brief rather than “briefly”

[1] Improving Graph Neural Network Expressivity via Subgraph Isomorphism Counting, IEEE Transactions on Pattern Analysis and Machine Intelligence
[2] Giannis Nikolentzos, George Dasoulas, and Michalis Vazirgiannis. k-hop graph neural networks.

---

### Official Review · Reviewer_qXEh · 2022-10-20

**Overall Score:** 8
**Confidence:** 4

**Review:**

**Summary**:

This paper proposes a "subgraph-aware" variant SaWL of the Weisfeiler Leman isomorphism test (WL) and a corresponding GNN, which provably can distinguish more graphs, i.e., is more expressive. The also authors also discuss an efficient instantiation of their proposed variant with only small computational overhead compared to WL using the color histogram of the $k$-hop subgraph of each node (complementing the usual WL subtrees of depth $k$). They achieve competitive performance on standard benchmark tasks.

**Main review**: *Straightforward novel variant of WL speeding up earlier work, not well written & some missing related work.*

While the main idea is nice and simple, the paper is difficult to follow because of the many grammar mistakes and the writing style. For example, a phrase like "subgraph-aware GNNs" can be easily mistaken with GNNs that use subgraph counts as additional features, instead of something like nested GNNs (what the author mean). While this is a rather incremental paper, I think it could be interesting for the LoG community, as it proposes a novel efficient message passing scheme (with expressiveness > 1-WL) and gets rid of the need to run linearly many base GNNs on all subgraphs (as is the case for e.g., nested GNNs). If the authors would improve their writing and make their ideas more easily understandable, I am recommending acceptance. Currently, I am leaning slightly towards acceptance, but can understand if it gets rejected (not too strong novelty, not well written).

**Missing related work**:

Some related work exists which was neither discussed not used in the empirical evaluations. Why? E.g.,

* ESAN of Bevilacqua et al., ICLR 2022 (very much related idea)
* there are many papers with (rooted) subgraph counts as additional node features (e.g., ..), (e.g., Bouritsas, et al. "Improving graph neural network expressivity via subgraph isomorphism counting." PAMI 2022).

**Questions**:

* What do you mean with "exponentially higher complexity like other [..] GNNs in the abstract? Most practical GNNs, also the ones referenced (e.g., nested GNNs), sometimes have a higher computational complexity but still only polynomially higher (or even just linearly). Also, the ones counting subgraphs are usually restricted to subgraphs of constant size and thus only yield polynomial additional overhead.
* As you mention Graph kernels in the appendix: Have you tried using the SaWL color histograms directly as a graph kernel (similarly to the well-known WL graph kernel) together with a (e.g., linear or RBF) SVM? This might be an interesting baseline compared to the proposed GNNs.

**Further minor remarks**:

* typo: references to Figure 1(c) and Figure 1(b) appears to be swapped in the main text.
* use a different font for S (the S operator). e.g., $S$, or $\texttt{S}$ to distinguish it from normal text. Also, in equation (3) the S operator is just called $\phi^{(h)}$. Why are two different names/symbols used?
* line 76: WL stops after a fixed number of iterations or as soon as the two graphs have different colors histograms. Then completely true: WL also stops if in one iteration nothings changes, i.e., the stable coloring is reached.
* if the "WL encoder" (line 97) merely executes WL, why do you need a new name for it?
* line 333: high-dimensional ("al") missing
* line 335: breifly --> brief
* line 338: nodes tuple --> node tuples
* line 39: "by its' rooted" --> "by its rooted".
* abstract: first sentence "the missing before "Weisfeiler-Lehman". "for graph isomorphism test" --> testing.
* language: often the indefinite article "a" is either missing or should be used instead of definite article "the" (e.g., "For graph $G$" --> For a graph $G$. "denotes the multiset" --> denotes a multiset).

------

**rebuttal:**

I've raised my score from weak accept to accept, see explanation in the comment to the authors below

---

### Official Review · Reviewer_nLWy · 2022-10-20

**Overall Score:** 1
**Confidence:** 5

**Review:**

**Contribution**: the paper proposed a modified subgraph based graph neural network. The author discussed the efficiency to existing subgraph based GNNs.

**Strong points**: the author's written is easy to follow. The visualization is good.

**Weak points**:
1. No much novelty. This is kind of incremental work with small modification to existing subgraph based GNN. Also, the proposed method is actually a special case of the paper [How Powerful are K-hop Message Passing Graph
Neural Networks, 2022 NeurIPS].
2. No much contribution in theory.

**Recommendation**: Clear Rejection.

---

### Official Review · Reviewer_aZix · 2022-10-22

**Overall Score:** 6
**Confidence:** 3

**Review:**

## Summary

The paper deals with the problem of supervised learning on graph structured input data. Graph Neural Networks (GNNs) which are commonly used for this task are limited by 1-WL graph isomorphism test in terms of their expressive power. This problem has been well studied recently with multiple works proposing to improve the expressive power of GNNs. One such technique is taking into consideration the subgraphs around every node. Those methods try to capture subgraph-level information which further improves the expressivity of GNNs. However, the computational complexity grows exponentially with the size of subgraphs.

The authors of this paper propose a version of such methods capturing subgraph information. However, the computational complexity does not blow up in this case. They propose Subgraph-aware WL (SAWL) and its neural counterpart SAGNN which pools the nodes in the subgraphs after each WL iteration with the help of an operator called “S operator”. Notably, the size of the subgraph captured grows in each iteration. Experiments are shown to provide evidence of the proposed approach along with some theoretical results suggesting more expressivity of SAWL.

## Strengths:
1. The paper addresses an important problem of finding efficient ways of improving the expressivity of the GNNs. It focuses on finding subgraph patterns around each nodes to distinguish different graph structures.
1. Novelty: Although, if we look carefully, this is not something completely new, there is some degree of insights.
1. The proposed method can clearly distinguish graphs beyond 1-WL. This is clear from the examples.
1. Experimental results on some datasets are good.
1. The paper itself is partly well-written and can be improved in some places.

## Weaknesses:
1. Although the paper claims to capture the subgraph information, it seems to me that it is not doing that. At least what other papers do when they capture the subgraph information is not what is done in the paper. SAWL simply takes the node aggregation of multi-hop neighbours and pools them after each iteration of 1-WL. This is more like capturing multi-hop information rather than capturing subgraph information. Even the example that is shown in Figure 2, is only able to distinguish after the second iteration when two-hop info is taken. However, for the method to say that it is capturing subgraph information, it should be able to distinguish in the first iteration itself since the two subgraphs in Figure 2 (d) are different. Clearly, the paper is incorrectly positioned as it seems. This decreases the novelty in the approach considerably. I believe, if we compare the expressive power of multi-hop GNNs like Mixhop (Abu-El-Haija et. al. (2019)) with SAWL, it will be same. Therefore, there is nothing additional introduced in this paper.
1. The above point is strengthened with Proposition-2 as well, since you use the distance to show SAWL is more expressive, not the subgraph info like GraphSNN.
1. The experimental results are clearly lacking. Since the aim of the paper is primarily to go beyond 1-WL GNNs, then it should present experiments where SAWL is able to identify graphs not identifiable by 1-WL GNNs on multiple datasets to support its primary claim.
For example, datasets like on Circular skip link (CSL), Strongly regular graphs (SR45) etc.. Most papers on the topic of expressive GNNs do exactly this. Specifically, it would strengthen the paper’s claim if there is comparison with higher-order K-GNNs like PPGNN (Maron et al.(2019) or recent PF-GNN (Dupty et al. (2021)). Subgraph based GraphSNN is compared only with real-world datasets and not EXP where it matters, since EXP is the only dataset beyond 1-WL reach.
1. Regarding computational complexity, it seems to me that it is incorrect to say it is same as 1WL. This is because, even though you can use the count of each subgraph when pooling the nodes over whole graph, you still need to precompute the counts. It seems the time complexity stays the same.
1. Ablation studies can be expanded. Most importantly since multi-hop info is the important factor, you can check the performance of the model at different iterations. This might give more insights.


To summarize, I find the idea interesting, however not very novel at the core-level of understanding. Additionally, the experiments are lacking in multiple ways and paper can be strengthened with additional ablation studies.


## References:

+ Abu-El-Haija, Sami, et al. "Mixhop: Higher-order graph convolutional architectures via sparsified neighborhood mixing." international conference on machine learning. PMLR, 2019.
+ Maron et al.(2019). "Provably powerful graph networks." Advances in neural information processing systems 32
+ Dupty et al. (2021) "PF-GNN: Differentiable particle filtering based approximation of universal graph representations." International Conference on Learning Representations.

---

### Meta-Review · Area_Chair_UTpR · 2022-11-09

**Confidence:** 4
**Recommendation:** Borderline and needs further discussi…

**Meta Review:**

The authors propose a simple enhancement of the 1-WL and corresponding neural architecture to overcome GNNs' limitation in expressivity. The main contribution is an efficient way to encode subgraph information during aggregation, provably overcoming the limitation of the 1-WL. Specifically, the authors first propose to run 1-WL and then aggregate labels in the $k$-hop neighborhood around each node. This simple enhancement achieves promising empirical performance.

All reviewers agree that the present work is quite incremental and that similar ideas have been explored before and that the authors should do a better job of discussing the key differences.  However, all but one reviewer liked the simplicity of the approach. However, they would like a clearer presentation and a more thorough experimental evaluation.

---

### Decision · Program_Chairs · 2022-11-23

Accept (Poster)